# In Vitro Interaction of Melanoma-Derived Extracellular Vesicles with Collagen

**DOI:** 10.3390/ijms24043703

**Published:** 2023-02-12

**Authors:** Roberta Palmulli, Enzo Bresteau, Graça Raposo, Guillaume Montagnac, Guillaume van Niel

**Affiliations:** 1Institut Curie, PSL Research University, CNRS, UMR144, Structure and Membrane Compartments, 75005 Paris, France; 2Inserm U1279, Gustave Roussy Institute, Université Paris-Saclay, 94805 Villejuif, France; 3Institute of Psychiatry and Neuroscience of Paris (IPNP), Université de Paris, INSERM U1266, 75014 Paris, France

**Keywords:** extracellular vesicles, collagen, ECM, melanoma

## Abstract

Extracellular vesicles are now considered as active contributors to melanoma progression through their capacity to modify the tumor microenvironment and to favor the formation of a pre-metastatic niche. These prometastatic roles of tumor-derived EVs would pass through their interaction with the extracellular matrix (ECM) and its remodeling, in turn providing a substrate favoring persistent tumor cell migration. Nevertheless, the capacity of EVs to directly interact with ECM components is still questionable. In this study, we use electron microscopy and a pull-down assay to test the capacity of sEVs, derived from different melanoma cell lines, to physically interact with collagen I. We were able to generate collagen fibrils coated with sEVs and to show that melanoma cells release subpopulations of sEVs that can differentially interact with collagen.

## 1. Introduction

Melanoma is the deadliest skin cancer and among the most aggressive solid tumors, mostly due to its high capacity of metastasizing in distant organs and its resistance to therapies [1]. Metastatic progression of melanoma is a multistep process that includes dysregulation of growth/apoptotic pathways, epithelial to mesenchymal transition and cell migration, immune system escape, basal membrane and stroma degradation, formation of metastatic niche, and angiogenesis [2]. Melanoma-derived extracellular vesicles (EVs) are active contributors to several steps of melanoma progression including angiogenesis and immune regulation but also remodeling of the tumor microenvironment [3].

The tumor microenvironment plays a major role during melanoma progression. It comprises various cell types, such as fibroblasts and immune cells, as well as non-cellular components, including soluble factors and the extracellular matrix (ECM) [4]. The ECM is a macromolecular network composed of collagens, proteoglycans/glycosaminoglycans, elastin, fibronectin, laminins, and other glycoproteins, which not only provide a physical scaffold in which cells are embedded but also contribute to cell differentiation, migration, and homeostasis [5]. Remodeling of the ECM structure occurs in both physiological and pathological conditions and is mediated by a variety of enzymes, including metalloproteases. In the context of cancer progression, ECM remodeling contributes to both cell transformation and migration.

Extracellular vesicles comprise heterogeneous groups of cell-derived membranous structures [6], some of which have been observed residing within the matrix [7] or seem capable of remodeling the ECM [8]. ECM remodeling by EVs would notably support the modification of the tumor microenvironment and the formation of a pre-metastatic niche [3]. Yet, it remains to be determined whether the observation of matrix-bound vesicles results from a passive retention or an active interaction of EVs with ECM components. The presence of ECM receptors (e.g., integrins); ECM components, such as fibronectin [9]; and active proteases at the surface of EVs [10] would support the capacity of EVs to bind and directly degrade the ECM [11,12,13] in the extracellular space.

Nevertheless, very few reports have investigated the capacity of EVs to physically interact with collagen, the most abundant fibrous component of the ECM [5]. In this study, we aimed to determine if melanoma-derived EVs were able to interact with collagen I.

## 2. Results

### 2.1. Melanoma Cells Secreted Subpopulations of EVs Containing ECM Components

To characterize small EVs (sEVs) released by melanoma cells, we chose three (human) melanoma cell lines with different metastatic potential: MNT-1, a pigmented cell line, not tumorigenic when injected in mice [14], and A375 and WM1716, which are non-pigmented, metastatic, melanoma cell lines. Small extracellular vesicles (sEVs) were isolated from sub confluent cultures of MNT-1, A375, or WM1716 by differential ultracentrifugation. The 100,000× *g* pellets were recovered and analyzed by Western blot together with their respective cell lysates (Figure 1A). As previously shown, MNT1 cells express ApoE, an inhibitor of melanoma progression [15], and secrete ApoE through sEVs [16]. On the contrary A375 and WM1716 cells do not express ApoE, and hence EVs released by these cell lines are negative for ApoE. All three cell lines express and secrete fibronectin, which can be co-recovered with sEVs. The presence of fibronectin on EVs has been previously reported and proposed to promote cancer cell migration through interaction with collagen [9]. The presence of sEVs was confirmed by transmission electron microscopy (Figure 1B). To further characterize sEVs, and confirm the physical association of fibronectin with EVs, 100,000 g pellets were loaded onto a bottom-up density gradient. Twelve fractions were recovered, their density was measured (Figure 1C), and each fraction was analyzed by Western blot (Figure 1D–F). MNT1 cells release two main subpopulations of EVs, corresponding to fraction 6/7 (density range 1.08–1.10 g/mL) and to fraction 9/10 (density range 1.12–1.17 g/mL) (Figure 1D). Contrary to CD63 and CD9, ApoE and fibronectin were specifically enriched in fraction 9/10 (Figure 1D), suggesting the presence of distinct subpopulations of sEVs. Similarly, A375-derived sEVs distributed in different fractions between 1.08 g/mL and 1.17 g/mL, with fibronectin being highly enriched only in fraction 9/10 (Figure 1E). Contrarily, WM1716 distributed in a single fraction (fraction 6, density of 1.08 g/mL), with this fraction being also enriched in fibronectin (Figure 1F).

### 2.2. Melanoma Derived-EVs Physically Interacted with Collagen

EV interaction with ECM has been investigated using matrix monomers [17] but never with structured ECM. In order to test the capacity of melanoma EVs to interact with ECM in vitro, we chose collagen type I, a fibrillar collagen widely spread in the body and a main component of the dermis, the first tissue that is invaded by melanoma cells [5]. To visualize the interaction of EVs with collagen, 100,000× *g* EV pellets (that comprise both subpopulations of sEVs) were incubated with a thin layer of collagen I, obtained by direct gelation on glass coverslips. The samples were then fixed, washed to remove unbound material, labeled for CD63 and CD9, and processed for conventional electron microscopy. We observed EVs positive either for CD63 or CD9 that are physically associated with fibrils of collagen (Figure 2A).

To obtain a quantitative measurement of the EV–collagen interaction, we took advantage of a recently developed protocol used to generate EGF- or LDL-decorated collagen fibers [18]. We first incubated 100,000× *g* EV pellets (that comprises both subpopulations of sEVs) with pre-polymerized collagen fibrils and performed a pull-down of these collagen fibrils. Collagen fibrils were pulled down by a low-speed centrifugation that allowed for the pull-down of collagen-associated EVs, while unbound EVs remained in solution. Western blot analysis revealed the presence of MNT1 sEV CD9 and CD63 markers in the pull-down, indicative of sEVs bound to collagen fibrils (Figure 2B). As the control, a pull-down assay was performed with collagen fibrils incubated with PBS instead of EVs (Figure 2B,C) and, as expected, no signal for CD63 or CD9 was detected. When we quantified the amount of MNT-1 sEVs that were pulled down (expressed as % of input EVs), no differences were observed between CD63 and CD9 signals (Figure 2B). Similarly, A375-derived EVs were bound to collagen (Figure 2C). In this case, we observed that the fraction of CD9-positive EVs associated with collagen was higher than the fraction of collagen-bound CD63 positive EVs (Figure 2C). Surprisingly, ApoE positive sEVs were not found in the pull-down, suggesting that ApoE containing EVs might not interact with collagen (Figure 2B). Moreover, fibronectin was found in the pull-down, suggesting that fibronectin containing EVs would interact with collagen. Nevertheless, fibronectin was also found when the assay was performed with collagen fibrils incubated with PBS, suggesting a possible presence of fibronectin contaminant in the purchased collagen I and hence making any observation regarding fibronectin inconclusive (Figure 2D).

### 2.3. Melanoma sEV Subpopulations Interacted with Collagen

The presence of different EV subpopulations in the supernatant of MNT-1 and A375 cells (Figure 1D–F) and the preferential pull-down of CD9 EVs over CD63 EVs suggested that each subpopulation could display different avidity for collagen fibrils. We then performed a pull-down assay using the fractions recovered upon density gradient.

Pull-down of MNT1 EV subpopulations revealed that mainly the fractions 6/7 interacted with collagen (Figure 3A), in line with the interaction of CD9- and CD63-positive EVs with collagen fibrils (Figure 2A,B). On the contrary, the ApoE and fibronectin containing fraction 9/10 was not pulled down with collagen (Figure 3A), in line with the absence of ApoE-containing EVs in the pull-down assay (Figure 2B). This was particularly evident for CD63, while a very low amount of CD9-positive EVs was still found (around 2% of the input) (Figure 3A). In the case of A375 EVs, both fraction 6/7 and fraction 9/10 were found to interact with collagen (Figure 3B). In both cases, a higher amount of CD9-positive EVs were pulled down, compared to CD63-positive EVs (Figure 3B). Finally, a pull-down assay was performed with the only EV-containing fraction released by WM1716 cells (Figure 3C). In this case, the association of CD63- or CD9-positive EVs with collagen was also observed, again with a majority of CD9-positive vesicles being pulled down (Figure 3C). Finally, the presence of EV-decorated collagen fibrils in the pull-down was also confirmed by electron microscopy (Figure 3D).

## 3. Discussion

In this study, we characterized small EVs secreted by three different melanoma cell lines, and we tested their capacity to interact with collagen, the most abundant component of the ECM.

Using an approach that combined ultracentrifugation and a bottom-up density gradient, we observed that melanoma cell lines released multiple subpopulations of sEVs, as previously reported [14,19]. We identified subpopulations of EVs that contained potential interactants of the ECM such as fibronectin and ApoE, confirming previous studies [9,14,20].

Using electron microscopy and a pull-down assay, we reported for the first time the capacity of melanoma sEVs to physically interact with fibrillar collagen I. While a previous report used flow cytometry to test the interaction of rat pancreatic cancer EVs with single ECM components, including collagen I [17], another report used an electron microscopy-based approached to show that EVs released by activated neutrophils physically interact with collagen I fibers and that they can degrade collagen, contributing to the pathogenesis of chronic obstructive pulmonary disease (COPD), a disease of lung ECM remodeling [21]. In addition, a recent study showed that human placental MSC-derived EVs were able to bind collagen-coated surfaces and that EVs could be conjugated to a collagen-binding peptide to increase their affinity for collagen binding [22]. Hence, EVs appear to interact with the ECM, and this interaction would likely contribute to ECM remodeling during different pathological processes.

Our pull-down assays suggested that CD9-positive EVs bind more efficiently to collagen than CD63-positive EVs. Although both CD63 and CD9 are common markers of sEVs, they are enriched in different subpopulations of sEVs that seem to originate from the endosomal system and the plasma membrane, respectively [23,24]. CD63 and CD9 EVs are enriched in a different subset of proteins that could affect their function, including their capacity to interact with ECM components.

Among the proteins that may influence the interaction of melanoma-derived EVs with collagen, we found fibronectin on EVs derived from the three cell lines, confirming previous studies [9,14]. Fibronectin is a component of the ECM that binds collagen [25] and might promote destabilization of collagen fibers, suggesting a role for fibronectin in collagen remodeling [26]. However, fibronectin was only present in the fraction 9–10 of MNT-1-derived EVs that poorly interacted with collagen, suggesting that the presence of fibronectin on sEVs might be dispensable for EV-collagen interaction.

In this study, we selected the non-tumorigenic pigmented melanoma cell line MNT-1 that expresses ApoE, a positive regulator of melanocyte pigmentation [16] and a negative regulator of melanoma progression [15]. Using our pull-down assay, we observed that the ApoE-containing EV subpopulation was not able to interact with collagen I. It cannot be excluded that, in vivo, ApoE EVs would interact with other components of the ECM such as heparan sulphate proteoglycans (HSPGs), which are known receptors of ApoE [27], as suggested by a previous observation that ApoE could bind HSPGs present in hepatocyte-derived ECM [20].

We reported here that melanoma-derived EV subpopulations can physically interact with collagen I fibrils. This direct interaction fuels the increasing importance of EVs in tumoral microenvironment remodeling. Binding of EVs with the fibrillar matrix would favor the action of metalloproteases [10,11,12] present at the surface of EVs and the remodeling of the ECM. In addition to a direct role of EVs in modulating ECM structure, the binding of EVs to the ECM could also impact their diffusion in the tumor microenvironment and therefore impact the capacity of EVs to interact with the different cells that constitute the tumor microenvironment, hence modulating EV functions.

This interaction also supports the hypothesis that matrix associated tumoral EVs would regulate the directional migration of tumor cells in vivo [9]. Beyond their role in tumor progression, our observations are also important to support the role of EVs in physiological matrix remodeling such as bone matrix regulation [28].

Our protocol of pull-down assay was developed from a previous assay used to probe the interaction of growth factor (EGF) or LDL with collagen fibers [18]. As shown here, this protocol could be transposed to other EV subpopulations of interest (e.g., EVs from different pathological stages, or EVs bearing or not specific collagen receptors such as integrins) and provide a basis to investigate the role of EVs in functional assays such as invasion assays and matrix remodeling. In addition, the functionalization of therapeutic EVs to bind collagen [22] opens new perspectives to use our conclusions and protocols in therapeutic assays.

While our study clearly established a direct interaction between melanoma EVs and collagen I fibrils, key points remain to be addressed. Previous reports of the presence of fibronectin and ApoE on EVs subpopulations [9,16] led us investigate their role in the interaction of EVs with collagen fibrils. However, we show here that subpopulations containing ApoE and fibronectin are likely not the major subpopulations interacting with collagen I. Moreover, contamination of collagen preparation with fibronectin prevented us from concluding any role of this protein in the observed interaction. Further studies are required to investigate the role of other EV proteins such as metalloproteases and especially integrins, given their contribution to organotropic metastasis [29] and tumor growth [30]. Proteomic analysis has been previously performed by Lazar et al. [14] at least for MNT-1 and A375L cell lines identifying integrins, but we were not able to detect integrins by Western blot analysis. Therefore, it would be of interest to profile each subpopulation of EVs in terms of ECM (and ECM- binding) components with a particular focus on integrins. Finally, it would also be of interest to study the capacity of EVs to interact with other types of collagens (e.g., collagen VI, a main component of basement membranes) and other ECM components.

## 4. Materials and Methods

### 4.1. Cell Culture

A375 cells were maintained in DMEM supplemented with 10% FBS and penicillin–streptomycin. MNT-1 cells were maintained in DMEM supplemented with 20% FBS, 10% AIM-V medium, sodium pyruvate, nonessential amino acids, and penicillin–streptomycin. WM1716 was maintained in RPMI supplemented with 10% FBS and penicillin–streptomycin. All cell lines were maintained in culture and used for experiments only up to 10 passages upon thawing. For sEV recovery, cells were maintained in the same medium, depleted for bovine EVs (obtained by overnight ultracentrifugation at 100,000× *g*). All reagents were from Gibco^TM^, ThermoFisher Scientific, Saint Herblain, France.

### 4.2. Antibodies and Reagents

Antibodies and their sources were as follows: anti-CD63 (ab23792, Abcam, Paris, France) (1:200 dilution for WB and EM), anti-CD9 was a kind gift of Eric Rubinstein (Inserm, U935, Villejuif, France) (dilution 1:500 for WB and 1:200 for EM), anti-ApoE (ab52607 or ab1906, Abcam, Paris, France) (dilution 1:500 for WB), anti-fibronectin (F3648, Sigma-Alrich, Merck, Saint Quentin Fallavier, France) (dilution 1:1000 for WB), anti-collagen (ab292, Abcam, Paris, France) (dilution 1:1000 for WB), horseradish peroxidase (HRP)-conjugated goat polyclonal antibodies to rabbit IgG and to mouse IgG (Abcam, Paris, France) (dilution 1:10000 for WB), protein A conjugated to 10 nm gold particles (Cell Microscopy Center, Utrecht University Hospital, Utrecht, The Netherlands) (PAG; dilution 1:50 for EM).

Rat tail collagen I (Catalog Number 354236) was from Corning, Sigma-Aldrich, Merck, Saint Quentin Fallavier, France.

### 4.3. EV Isolation

sEVs were prepared from conditioned media incubated for 48 h on sub-confluent cells (70–80% confluency), grown either in 75 cm^2^ or 150 cm^2^ flasks (10 × 10^6^ or 20 × 10^6^ cells per flask, respectively). Conditioned media were centrifuged at 300× *g* (15 min, 4 °C) and 2000× *g* (20 min, 4 °C) to remove cell debris. Next, the supernatant was centrifuged at 10,000× *g* (30 min, 4 °C), and sEVs were collected from the supernatant by centrifugation at 100,000× *g* for 60 min (4 °C, 45 Ti or 70 Ti rotor, Beckman Coulter, Brea, CA, USA). The pellet was washed in PBS (pH 7) by centrifugation at 100,000× *g* (60 min, 4 °C) and finally resuspended in PBS (pH 7).

Bottom-up density gradient was performed as previously described [31] with some modifications. Solutions of 5, 10, 20, and 40% iodixanol were made by mixing a homogenization buffer (0.25 M sucrose, 1 mM EDTA, 10 mM Tris-HCL, (pH 7.4)) and an iodixanol working solution, prepared by combining a working solution buffer (0.25 M sucrose, 6 mM EDTA, 60 mM Tris-HCl, (pH 7.4)) and a stock solution of OptiPrep^TM^ (60% (*w*/*v*) aqueous iodixanol solution, Sigma-Aldrich, Merck, Saint Quentin Fallavier, France). The 100,000 g EV pellets were mixed with the bottom fraction to obtain a 40% fraction. The gradient was formed by layering 40%, 20%, 10%, and 5% solutions on top of each other in an open-top polypropylene tube (Beckman Coulter). The gradient was then centrifuged for 14 h at 100,000× *g* (4 °C, SW41 rotor, Beckman Coulter). A total of 12 gradient fractions of 1 mL were manually collected from the top of the gradient, diluted in PBS, and centrifuged for 1 h at 100,000× *g* (4 °C). The resulting pellets were resuspended in equal volumes of PBS (pH 7).

### 4.4. Western Blot

Cells were lysed directly in lysis buffer (20 mM Tris, 150 mM NaCl, 1% Triton X-100, 1 mM EDTA; pH 7.2) with protease inhibitor cocktail (Roche). Cell lysates, EVs, or pull-down pellets were incubated with sample buffer with or without 350 mM 2-mercaptoethanol (Sigma-Aldrich, Merck, Saint Quentin Fallavier, France) and incubated at 60 °C for 30 min. Equal volumes were loaded on 4–12% Bis-Tris gels (Nu-PAGE, Invitrogen), and proteins were transferred on nitrocellulose membranes (GE Healthcare). Membranes were blocked in PBS/0.1% Tween (PBS/T) with 5% nonfat dried milk and incubated with indicated primary (overnight at 4 °C) and secondary antibodies (1 h at room temperature) diluted in PBS/T-milk. Western blots were developed using the ECL SuperSignal West Pico or Dura (ThermoFisher Scientific, Saint Herblain, France). The presented immunoblots are representative of at least three independent experiments. Signal intensities from at least three independent experiments were quantified with Image J Fiji software (version 2.3.0). Graphs were prepared using GraphPad Prism version 9.5.

### 4.5. Electron Microscopy

For conventional electron microscopy, collagen I mix was prepared by mixing rat tail collagen I (final concentration 2.2 mg/mL) with 10 X PBS, 1 M HEPES, and H_2_O, and the pH was neutralized by adding 1 M NaOH. Collagen gelation was conducted on coverslips for 10 min at RT. sEVs were added on top of the collagen-coated coverslips and incubated for 20 min at RT. After a washing step in phosphate buffer, coverslips were fixed with a mixture of 2% PFA and 0.2% glutaraldehyde in 0.1 M phosphate buffer (pH 7.4), quenched with PBS/50 mM glycine, and processed for immunogold labeling using anti-CD63 or anti-CD9 antibody and PAG 10 nm as previously described [32]. Coverslips were then fixed with 2.5% glutaraldehyde in 0.1 M cacodylate buffer and processed for Epon (TAAB Laboratories Equipment) embedding and ultrathin sectioning. Sections were then contrasted with uranyl acetate and lead citrate, as previously described [32].

For EM analysis of EVs or of the pull-down of collagen fibrils incubated with EVs, EVs or pull down (prepared as described below) were spotted on formvar/carbon-coated copper/palladium grids, incubated for 20 min before fixation with PFA 2%/0.1 M phosphate buffer, and washing with water. Then, negative staining was performed using 0.4% uranyl acetate in methylcellulose.

The samples were analyzed with an 80 kV transmission electron microscope (Tecnai Spirit G2; Thermo Fischer, Eindhoven, The Netherlands) equipped with a 4k CCD camera (Quemesa, EMSIS, Münster, Germany).

### 4.6. Preparation of EV-Decorated Collagen Fibers for Pull-Down Assay

Collagen I mix was prepared as above. EV-decorated collagen fibers were generated using a protocol previously used to generate EGF- or LDL-decorated collagen fibers [18]. Collagen gelation was conducted in low-retention tubes for 10 min at RT and stopped by adding cold PBS and keeping samples in ice. Collagen gels were then sonicated on ice 3 times for 10 s, with 10 s breaks in between, at 50% amplitude. EVs, isolated as described above, were added to the collagen gels, and the samples were incubated for 2 h 30 min at RT. One volume of PBS was added, and samples were sonicated on ice 12 time for 10 s, with 10 s breaks in between, at 50% amplitude. Collagen fibrils (and bound EVs) were pulled down by centrifugation at 2150× *g* for 1 h at 4 °C. The supernatant (unbound fraction) was recovered, and the volume was reduced using Microcon Centrifugal Filter Devices (Millipore, Merck, Molsheim, France) according to the manufacturer’s instructions. Pellets (pull down) were washed with cold PBS, resuspended in PBS, and processed for WB as described above. The same volume of EVs used for the pull-down was analyzed by WB (input). As a control, pull-down assay was performed with collagen incubated with PBS instead of EVs (coll + PBS).

## Figures and Tables

**Figure 1 ijms-24-03703-f001:**
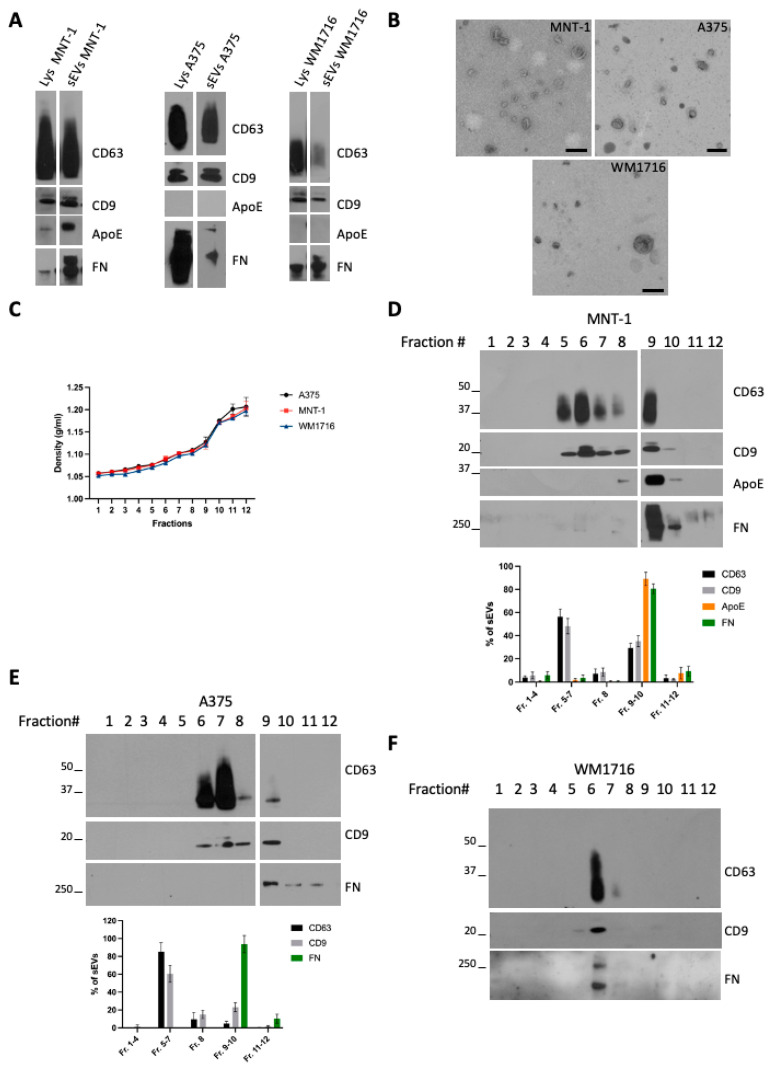
(**A**) Western blot analysis of cell lysates and 100,000× *g* sEV pellets from the melanoma cell lines MNT-1, A375, and WM1716. (**B**) TEM analysis of sEVs from the melanoma cell lines MNT-1, A375, and WM1716. Scale bar = 200 nm. (**C**) Density values of the fractions recovered from a bottom-up density gradient of 100,000× *g* sEV pellets from A375, MNT-1, and WM1716 cells (mean ± SEM, 3 independent experiments). (**D**) Western blot analysis of a bottom-up density gradient of 100,000× *g* sEV pellet from MNT-1 cells. Quantification of EV markers’ distribution in fractions collected from the density gradient (mean ± SEM, 3 independent experiments). (**E**) Western blot analysis of a bottom-up density gradient of 100,000× *g* sEV pellet from A375 cells. Quantification of EV markers’ distribution in fractions collected from the density gradient (mean ± SEM, 3 independent experiments). (**F**) Western blot analysis of a bottom-up density gradient of 100,000× *g* sEV pellet from WM1716 cells.

**Figure 2 ijms-24-03703-f002:**
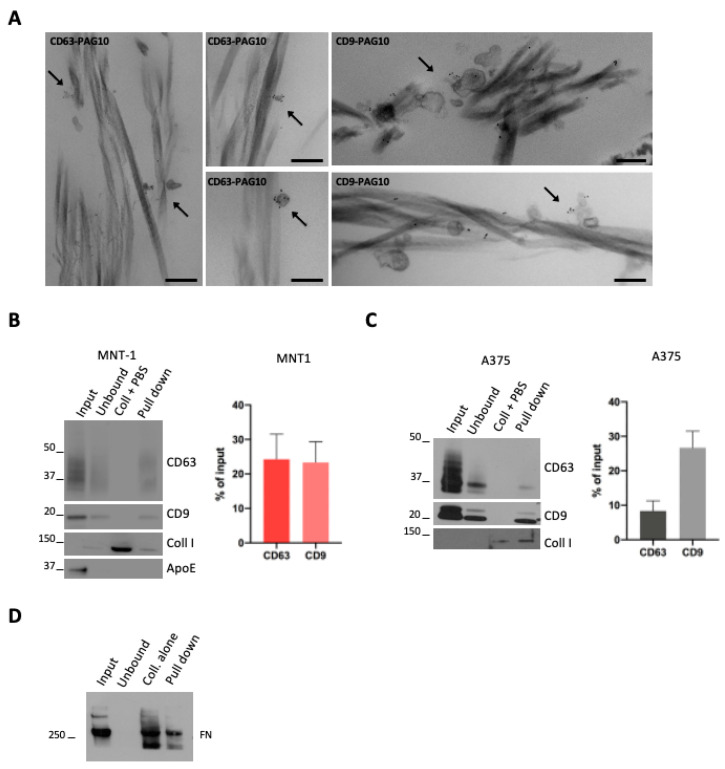
(**A**) A375 sEVs were incubated with collagen, pre-stained with CD63 or CD9 (PAG-10), and analyzed by conventional EM. Arrows indicate stained EVs associated with collagen. Scale bar = 200 nm. (**B**,**C**) WB analysis of a pull-down of collagen fibrils incubated with sEVs. Quantifications of the percentage of CD63- or CD9-positive vesicles that were pulled down with collagen are shown (mean ± SEM, 3 independent experiments). (**D**) WB analysis of a pull-down of collagen fibrils incubated with sEVs. A representative blot for fibronectin is shown.

**Figure 3 ijms-24-03703-f003:**
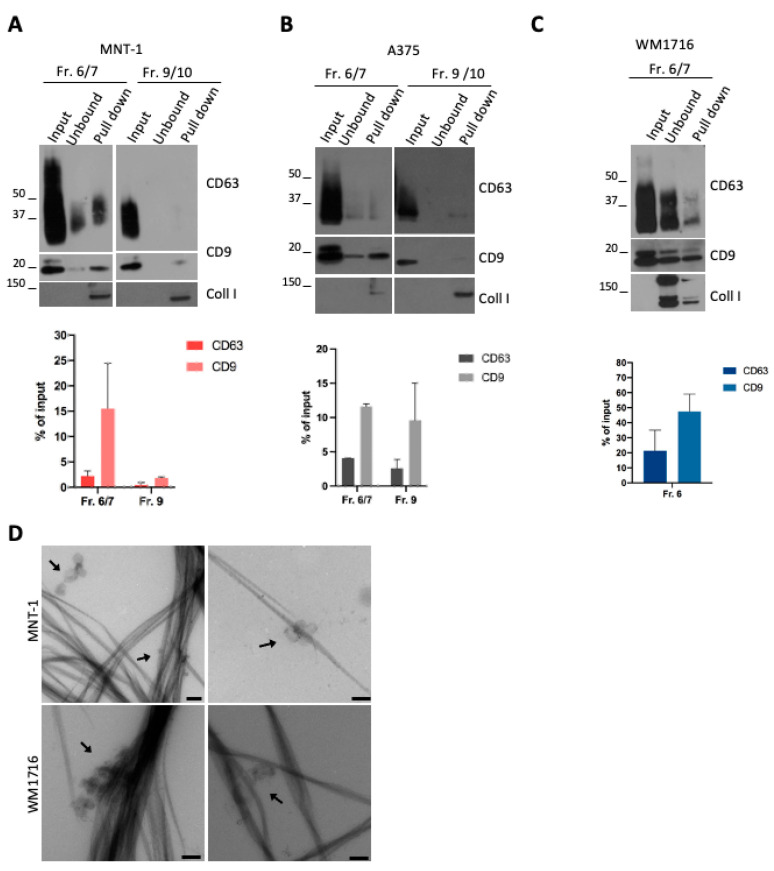
(**A**) WB analysis of a pull-down of collagen fibrils incubated with MNT-1 sEV fractions recovered from a bottom-up density gradient. Quantifications of the percentage of CD63- or CD9-positive vesicles that were pulled down with collagen are shown (mean ± SEM, 3 independent experiments). (**B**) WB analysis of a pull-down of collagen fibrils incubated with A375 sEV fractions recovered from a bottom-up density gradient. Quantifications of the percentage of CD63- or CD9-positive vesicles that were pulled down with collagen are shown (mean ± SEM, 3 independent experiments). (**C**) WB analysis of a pull-down of collagen fibrils incubated with WM1716 sEV fraction recovered from a bottom-up density gradient. Quantifications of the percentage of CD63- or CD9-positive vesicles that were pulled down with collagen are shown (mean ± SEM, 3 independent experiments). (**D**) EM micrograph of a pull-down of collagen fibrils incubated with MNT-1 sEVs or WM1716 sEVs fraction 6/7. Scale bar = 200 nm.

## Data Availability

The data presented in this study are contained within the article. We have submitted all relevant data of our experiments to the EV-TRACK knowledgebase (EV-TRACK ID: EV230014) “https://evtrack.org/” (accessed on 10 February 2023) [33].

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
