# Peer review of "In Vitro Interaction of Melanoma-Derived Extracellular Vesicles with Collagen"

_ijms, 2023, doi:10.3390/ijms24043703_

Round 1

Reviewer 1 Report

In this manuscript, Roberta Palmulli et al. describe a series of experiments on the in vitro interaction of extracellular vesicles with collagen in melanoma cell lines.

The paper shows interesting data, but in my opinion, some modifications are needed.

1.- In Figure 1, the authors evaluated markers expressed in EVs and microscopic analysis to evaluate the interaction between collagen and EVs in Figure 2D, it would be interesting to make a section showing only the characterization of the EVs obtained.

2.- In Figure 2D, the authors show a microscopic analysis where they show the presence of collagen fibrils that interact with the EVs of the MNT-1 cell line, why do they only show the analysis of one cell line and not of the A375 cell line that has greater metastatic potential, and the central theme is the interaction of the EVs with collagen, then I consider that it is important to add the images of the other 2 cell lines. It is also suggested to study the presence of other proteins capable of interacting with the ECM and that is present on the surface of the EVs, or to perform a proteomic analysis to identify the proteins present in these EVs that could mediate this EV-collagen interaction.

3.- How do the authors ensure that the microscopic analysis showing the interaction of collagen fibrils with EVs is an "interaction" and not an artifact of the technique?

4.- The authors should specify in figure 2E in which cell line they performed the experiment and what it means that they found fibronectin expression when treated with PBS compared to pull down.

MINOR POINTS

 1.- The authors should be precise in the methodology with the catalog number of the reagents used, as well as specify how they obtained the 12 fractions shown in Figure 1C-1E, for the reproducibility of the study.

2.- In the materials and methods section, the authors do not mention the number of cells that were cultured on the plates from which the conditioned medium for obtaining the EVs was obtained. In addition, the area of the cell culture vessel is not mentioned.

 3. The authors should also include relevant information on the number of passages of the cell lines on which the experiments shown were performed for reproducibility purposes.

 4. The authors should include in the discussion section the limitations of the study, given that the main objective of the work was to demonstrate the interaction of EVs secreted by melanoma cells with collagen I. However, the key players (EV proteins) in this interaction were not fully described. These points should be addressed as limitations and future research or perspectives.

 5.- They should make an exhaustive revision of the syntax.

Reviewer 2 Report

 The communication titled "In vitro interaction of extracellular vesicles with collagen" by Palmulli et al. is concise and well structured and clarifies how melanoma cells release subpopulations of sEV that can differentially interact with collagen.

My comments aimed at perfecting a job already well done concern the improvement of the resolution of some images (Fig. 1B and 1E).

Given the interest in this research field with related publications such as: doi: 10.3390/ijms23063389, doi: 10.7150/thno.70448, I ask you to improve the bibliography which is not updated.

And finally, I ask you to enter the extracellular vesicle information into the EV-TRACK database (https://evtrack.org/)

Reviewer 3 Report

The manuscript is well written. However, the authors should address the following queries:

Major Comments:

1.     The title should be specified (such as in which cancer).

2.     Please discuss the importance of the study.

3.     Does the EVs contain any inflammatory cytokines?

4.     Why did the authors focus only on Collagen 1 and not the other types of collagens?

5.     Why didn’t the authors consider the interaction of EVs with other important ECM proteins like fibronectin?

6.     Did the authors find any role of the EVs on the metastatic nature of the particular cell line.

7.     Matrix metalloproteinases (MMPs) are crucial in ECM regulation. Therefore, MMPs can influence the interaction of EVs with the ECM like collagen 1. Do the authors have any data regarding this interaction? The authors should consider this aspect seriously.

Minor Comments:

1.     Some error bars in Fig. 3 are too high!

Reviewer 4 Report

Interesting work, but some elements need to be completed.

1.      Please organize sections 4.1. -4.6. in such a way that each of the listed reagents is assigned the company from which it was purchased (company name, city, country). This may be questionable at the moment. Please do not put the company name, antibody clone number and dilutions used in the same bracket.

2.      The manuscript lacks EV sample characterization using Transmission electron microscopy (TEM) and nanoparticle tracking analysis (NTA) to evaluate sample purity/contamination and vesicle size. Such information should at least be included in the supplementary data.

3.      How much protein (in micrograms) was added to the wells of 4-12% Bis-Tris gels.

4.      Please add in section 4.4 information on Ab incubation times.

5.      There is no information on the manner and scope of statistical analysis of the obtained results.

6.      The method of providing cited references is inconsistent with the requirements in the Author guide. This will need improvement.

Reviewer 5 Report

Palmulli et al. show that some subpopulations of small EVs isolated from three different melanoma cell lines interact with collagen in vitro. This conclusion is based on experimental data obtained from pull-down assays and electron microscopy analysis.

1. In general the manuscript is well written and supported by experimental data. My major concern is the quantification of western blot analysis. The signals obtained are really different, some very strong, the others barely detectable, which means that the strong signals (for instance input for CD63) is clearly out of linear range and the resulting numbers (% of input) do not adequately reflect the results. Second, statistics should be performed to confirm that these differences are statistically significant (some error bars are very wide).

2. The results shown in Fig. 3B are confusing. The western blot images of CD9 pull-down from Fr. 9/10 in Fig. 3A and B are very similar (CD9 signal is barely visible), but the quantification shows big differences (10% of input in Fig. 3B and only 1-2% in Fig. 3A). Based on the image of CD9 on Fig. 3B, CD9 is not pulled down from Fr. 9/10 and the conclusion on lanes 141-142 “In the case of A375 EVs, both fraction 6/7 and fraction 9/10 were found to interact with collagen” is not correct. In addition, Fr 9/10 images of Fig. 3B are different from original images – what is input and what is pull-down? In the original images the strongest bands are always in input fraction, but in Fig. 3B, the strong band of collagen is in pull-down fraction. What is correct?

Round 2

Reviewer 1 Report

The authors addressed all comments and suggestions, the manuscript is ready for publication.

kind regards

Author Response

We thank the reviewer for their comments that contributed to improve our manuscript.

Reviewer 3 Report

The manuscript is reasonably improved now and can be accepted for publication after checking it thoroughly for any grammatical errors and typos.

Author Response

We thank the reviewer for their comments and their contribution to improve our manuscript. The manuscript has now been checked for grammatical errors and typos.